# Recrystallization of Triple Superphosphate Produced from Oyster Shell Waste for Agronomic Performance and Environmental Issues

Somkiat Seesanong [1], Chaowared Seangarun [2], Banjong Boonchom [2,3,*], Chuchai Sronsri [2], Nongnuch Laohavisuti [4], Kittichai Chaiseeda [5,*] and Wimonmat Boonmee [6]

[1] Department of Plant Production Technology, School of Agricultural Technology, King Mongkut's Institute of Technology Ladkrabang, Bangkok 10520, Thailand; ksesomki@kmitl.ac.th

[2] Advanced Functional Phosphate Material Research Unit, Department of Chemistry, School of Science, King Mongkut's Institute of Technology Ladkrabang, Bangkok 10520, Thailand; 62605057@kmitl.ac.th (C.S.); chuchai_sronsri@hotmail.com (C.S.)

[3] Municipal Waste and Wastewater Management Learning Center, School of Science, King Mongkut's Institute of Technology Ladkrabang, Bangkok 10520, Thailand

[4] Department of Animal Production Technology and Fishery, School of Agricultural Technology, King Mongkut's Institute of Technology Ladkrabang, Bangkok 10520, Thailand; nongnuch.la@kmitl.ac.th

[5] Organic Synthesis, Electrochemistry and Natural Product Research Unit (OSEN), Department of Chemistry, Faculty of Science, King Mongkut's University of Technology Thonburi, Bangkok 10140, Thailand

[6] Department of Biology, School of Science, King Mongkut's Institute of Technology Ladkrabang, Bangkok 10520, Thailand; wimonmat.bo@kmitl.ac.th

* Correspondence: banjong.bo@kmitl.ac.th (B.B.); kittichai.cha@kmutt.ac.th (K.C.); Tel.: +66-2329-8400 (B.B.)

**Abstract:** Calcium dihydrogen phosphate monohydrate ($Ca(H_2PO_4)_2 \cdot H_2O$) (a fertilizer) was successfully synthesized through a recrystallization process using prepared triple superphosphate (TSP) derived from oyster shell waste as the starting material. This bio-green, eco-friendly process to produce an important fertilizer can promote a sustainable society. The shell-waste-derived TSP was dissolved in distilled water and kept at 30, 50, and 80 °C. Non-soluble powder and TSP solution were obtained. The TSP solution fractions were then dried, and the recrystallized products (RCP30, RCP50, and RCP80) were obtained and confirmed as $Ca(H_2PO_4)_2 \cdot H_2O$. Conversely, the non-soluble products (NSP30, NSP50, and NSP80) were observed as calcium hydrogen phosphate dihydrate ($CaHPO_4 \cdot 2H_2O$). The recrystallized yields of RCP30, RCP50, and RCP80 were found to be 51.0%, 49.6%, and 46.3%, whereas the soluble percentages were 98.72%, 99.16%, and 96.63%, respectively. RCP30 shows different morphological plate sizes, while RCP50 and RCP80 present the coagulate crystal plates. X-ray diffractograms confirmed the formation of both the NSP and RCP. The infrared adsorption spectra confirmed the vibrational characteristics of $HPO_4^{2-}$, $H_2PO_4^{-}$, and $H_2O$ existed in $CaHPO_4 \cdot 2H_2O$ and $Ca(H_2PO_4)_2 \cdot H_2O$. Three thermal dehydration steps of $Ca(H_2PO_4)_2 \cdot H_2O$ (physisorbed water, polycondensation, and re-polycondensation) were observed. $Ca(H_2PO_4)_2$ and $CaH_2P_2O_7$ are the thermodecomposed products from the first and second steps, whereas the final product is $CaP_2O_6$.

**Keywords:** $Ca(H_2PO_4)_2 \cdot H_2O$; triple superphosphate; fertilizer; recrystallization; oyster shell; thermal decomposition

## 1. Introduction

Calcium-phosphate-based materials, i.e., tetracalcium phosphate ($Ca_4(PO_4)_2O$), tricalcium phosphate ($Ca_3PO_4$), calcium dihydrogen phosphate monohydrate ($Ca(H_2PO_4)_2 \cdot H_2O$) or monocalcium phosphate monohydrate (MCPM), calcium hydrogen phosphate ($CaHPO_4$ or monetite), calcium hydrogen phosphate dihydrate ($CaHPO_4 \cdot 2H_2O$) or brushite), octacalcium dihydrogen phosphate pentahydrate ($Ca_8H_2(PO_4)_6 \cdot 5H_2O$), and decacalcium

hydroxide phosphate ($Ca_{10}(PO_4)_6(OH)_2$ or hydroxyapatite) have been thoroughly investigated [1,2] because they are stable, biocompatible, and similar to natural teeth and bone. Therefore, they have been used as bone substitutes in biomedical fields [2].

In the agricultural industry, calcium phosphate, an essential fertilizer, is called triple superphosphate (TSP) [3]. TSP fertilizer in the form of $Ca(H_2PO_4)_2 \cdot H_2O$ (or MCPM) provides a high concentration of phosphorus (P) (more than 40% of diphosphorus pentoxide, $P_2O_5$). It was widely used in the 20th century and became the most common phosphorus source used in many countries until the mid-1970s for plant development [4]. Phosphorus is one of the macronutrients employed for plant growth that plays an important role in many physiological and biochemical plant activities [5]. One of the advantages of feeding plants phosphorus is creating deeper and more abundant roots, early maturation, decreasing grain mold, increasing chlorophyll content in leaves, and positively affecting photosynthesis, resulting in the improvement of crop quantity and quality [6]. However, the agronomic performance of the fertilizer for plants is limited by its insolubility in water, which causes a very slow phosphorus delivery rate. To increase the soluble phosphorus contents and purity of the TSP fertilizer, it can be recrystallized to obtain soluble $Ca(H_2PO_4)_2 \cdot H_2O$. TSP is firstly dissolved in water, and a filtration technique is then employed to remove the insoluble fraction from the solution. When the dissolved cation and anion species are recrystallized, pure water-soluble $Ca(H_2PO_4)_2 \cdot H_2O$ can be recovered [7]. Owing to its high solubility, $Ca(H_2PO_4)_2 \cdot H_2O$ can deliver phosphorus at a higher rate [7]. Readily soluble phosphorus has the potential to supply phosphorus to plants immediately after application. When the fertilizers (i.e., $Ca(H_2PO_4)_2 \cdot H_2O$) are used, the moisture that exists in the soil is absorbed by $Ca(H_2PO_4)_2 \cdot H_2O$ (in both granular and powder forms). $Ca(H_2PO_4)_2 \cdot H_2O$ is dissolved, leading to the release of phosphorus fertilizer into the soil, followed by the uptake of plants for growth and development.

There are many synthesis methods in the production of calcium phosphates such as hydrothermal [4,6], microwave-assisted [8], precipitation [9,10], sol–gel [11], and electrochemical [12] techniques. The method used in each case depends on the desired type of morphology, structure, and chemical composition. In general, calcium phosphates have been synthesized from phosphoric acid ($H_3PO_4$) and commercial calcium ($Ca^{2+}$) sources [7]. However, using shells (one of the important wastes observed in many countries) as a raw $Ca^{2+}$-source material can reduce the cost of calcium phosphate production [13]. Oyster, one of the bivalve mollusks, can be found in coastal areas where brine water mixes with fresh water. Thailand is one of the oyster-rich countries. Eastern and southern parts of Thailand are areas where oysters can be found, and the two highest oyster production provinces are Chonburi (Eastern of Thailand) and Surat Thani (Southern of Thailand). *Crassostrea iredalei*, *Saccostrea cucullata*, and *Crassostrea belcheri* are the three major oyster species widely cultivated [14]. In 2019, Thailand's coastal aquaculture produced about 17,903 tons of oysters [15]. The results by Chilakala et al. [15] indicated that the major phase of raw oyster was $CaCO_3$ (92.6 wt% calcite and 2.71 wt% aragonite), which is an important source of calcium ($Ca^{2+}$). The X-ray fluorescence (XRF) results reported by Chilakala et al. [15] also demonstrated that the chemical composition of an oyster was mostly calcium oxide (CaO) (53.66 wt%). Furthermore, oxides of silicon ($SiO_2$, 0.45 wt%), aluminum ($Al_2O_3$, 0.12 wt%), iron ($Fe_2O_3$, 0.06 wt%), magnesium (MgO, 0.26 wt%), potassium ($K_2O$, 0.06 wt%), sodium ($Na_2O$, 0.55 wt%), titanium ($TiO_2$ < 0.01 wt%), manganese (MnO, 0.01 wt%), and phosphorus ($P_2O_5$, 0.16 wt%) were also observed [15].

Consequently, recycling waste materials (i.e., oyster shells) by applying a $Ca^{2+}$ source (in the form of $CaCO_3$) is beneficial for reducing waste and most useful for various industrial uses. The aim of this work is to recrystallize the soluble calcium phosphate ($Ca(H_2PO_4)_2 \cdot H_2O$) from the synthesized triple superphosphate (TSP) compound. Firstly, the oyster-shell waste was used as the precursor to prepare $CaCO_3$. This oyster-derived $CaCO_3$ then reacted with $H_3PO_4$, resulting in the formation of TSP. After that, the obtained TSP compounds were dissolved in deionized water at three different temperatures. The non-soluble part was filtrated, whereas the solution part was dried, resulting in the recrys-

tallization (formation) of $Ca(H_2PO_4)_2 \cdot H_2O$. This obtained $Ca(H_2PO_4)_2 \cdot H_2O$ exhibits the desired properties, namely high crystallinity, high purity, and high solubility. To confirm the formation of the target $Ca(H_2PO_4)_2 \cdot H_2O$ material, the physicochemical properties of the recrystallized products were identified and characterized by X-ray diffraction (XRD), Fourier transform infrared spectroscopy (FTIR), thermogravimetric/differential thermal analysis (TG/DTA), and scanning electron microscopy (SEM). In addition, the recrystallized yield and the solubility of $Ca(H_2PO_4)_2 \cdot H_2O$ were also calculated to demonstrate the influence of the operating condition on the properties of the synthesized $Ca(H_2PO_4)_2 \cdot H_2O$ compound. By successfully producing $Ca(H_2PO_4)_2 \cdot H_2O$, we have provided an efficient method to produce this valuable compound and also demonstrated a way to recycle oyster shell waste, which will lead to a higher income and a reduction in oyster shell waste to reduce environmental issues.

## 2. Materials and Methods

### 2.1. Preparation

Oyster shell waste was used as the starting material ($CaCO_3$ as $Ca^{2+}$ source) to prepare the triple superphosphate (TSP). After that, the oyster-shell-derived TSP compound was then used as the precursor to recrystallize $Ca(H_2PO_4)_2 \cdot H_2O$ (MCPM). Figure 1 shows the schematic diagram designed for MCPM preparation.

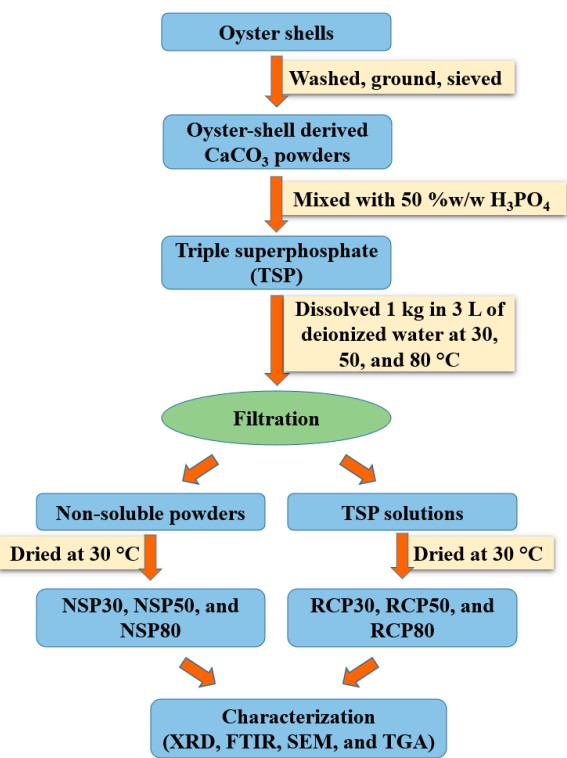

**Figure 1.** Schematic diagram designed for $Ca(H_2PO_4)_2 \cdot H_2O$ (MCPM) preparation using oyster-shell-derived TSP (triple superphosphate) as the precursor.

The waste was collected from a local shellfish market in Chonburi, Thailand, and was cleaned with sodium hypochlorite (NaOCl, saturated solution) until the shell's meat was completely destroyed. After the cleaning process, the shells were kept at 110 °C in an oven for 60 min. The dried shells were milled and then sieved by a 100-mesh sieve. Using this process, approximately 140 μm oyster-shell-derived $CaCO_3$ powders were achieved and used as a raw material. A concentration of 50% *w/w* $H_3PO_4$ prepared from 85% *w/w* industrial grade $H_3PO_4$ was also used as a reagent for the TSP preparation. The process described by Seesanong et al. [16] was used to prepare the TSP. A volume of 13.97 mL of 50% *w/w* $H_3PO_4$ was added to a beaker containing 10 g of shell-derived $CaCO_3$ powder.

The mixed suspension was stirred at 100 rpm until carbon dioxide ($CO_2$) gas was not evolved (approximately 30 min). The pale yellow-white product was obtained and exposed to ambient conditions for 12 h to dry. Equation (1) shows the chemical reaction between oyster-shell-derived $CaCO_3$ and $H_3PO_4$, and TSP ($Ca(H_2PO_4)_2·H_2O$) was then synthesized from this reaction.

$$CaCO_3(s) + 2H_3PO_4(aq) \rightarrow Ca(H_2PO_4)_2·H_2O(s) + CO_2(g) \tag{1}$$

Afterward, 1 kg of synthesized TSP was then dissolved in 3 L of deionized water and kept at three different temperatures, namely 30, 50, and 80 °C for 2 h. According to the hydrolyzed reaction, Equation (2), TSP was dissolved, while phosphoric acid ($H_3PO_4$) and calcium hydrogen phosphate ($CaHPO_4$) species were generated from this hydrolyzed reaction, which are the intermediates used to recrystallize and obtain MCPM ($Ca(H_2PO_4)_2·H_2O$).

$$Ca(H_2PO_4)_2·H_2O + H_2O \leftrightarrow H_3PO_4 + CaHPO_4 + 2H_2O \tag{2}$$

However, a fraction of TSP did not dissolve in water. Therefore, the resulting solution was filtered by filter paper using a Buchner funnel under reduced pressure. After filtration, the non-soluble powders on the filter paper were dried at 30 °C for 1 h, and the non-soluble powders processed at 30, 50, and 80 °C were labeled NSP30, NSP50, and NSP80, respectively.

In the case of the soluble fraction, the obtained TSP solutions were kept and dried at 30 °C for 3 h. Using this drying process, the recrystallized $Ca(H_2PO_4)_2·H_2O$ (MCPM) products were obtained. The recrystallized products prepared at 30, 50, and 80 °C were coded as RCP30, RCP50, and RCP80, respectively. The percent yields of the recrystallized products were calculated using Equation (3).

$$Yield(\%) = \frac{m_{obs}}{m_{theor}} \times 100 \tag{3}$$

where $m_{obs}$ is the mass of the synthesized $Ca(H_2PO_4)_2·H_2O$ powder from each recrystallized temperature and $m_{theor}$ is the mass of the theoretical $Ca(H_2PO_4)_2·H_2O$ (mass before recrystallization process).

### 2.2. Characterization

Dried powder of the recrystallized product was then ground by the ball mill technique at 60 rpm for 15 min under an ambient temperature. The solubility performance of the recrystallized samples was determined by adding 1 g of each sample into 100 mL of deionized water with the stirring mode at 60 rpm for 1 h. The resulting mixture was then filtered by a Buchner funnel, dried at 60 °C for 6 h, and weighed to determine the solubility value of the recrystallized samples.

The non-soluble powders (NSP30, NSP50, and NSP80) and the recrystallized products (RCP30, RCP50, and RCP80) were characterized by X-ray powder diffractometer (XRD, D8 Advance, Bruker AXS, Karlsruhe, Germany) to identify the structure, purity, and crystallinity of the prepared samples. Two theta ($2\theta$) angles ranged from 5° to 60° with the *Cu-K$_\alpha$* X-ray source. The selected scan speed was 1 s per step at a 0.01° increment. The International Centre for Diffraction Data (ICDD) database was used to compare and confirm the crystal structure and purity of samples obtained from experimental XRD results. Surface morphologies of samples were observed by a scanning electron microscope (SEM, FEI Quanta 250, Hillsboro, OR, USA) with energy-dispersive X-ray (EDX) mode. This EDX technique was used to evaluate the elemental compositions of samples. The gold-coated technique was used for sample preparation [17]. Infrared adsorption spectra were recorded on the Fourier transform infrared spectrophotometer (FTIR, Spectrum GX, PerkinElmer, Waltham, MA, USA). The measurement range is between 4000 and 370 $cm^{-1}$. A resolution of 2 $cm^{-1}$ was selected with 32 scans to decrease the noise signal [18]. To prepare the infrared adsorption sample, each non-soluble (NSP30, NSP50, and NSP80) and recrystallized (RCP30, RCP50, and RCP80) product was homogeneously mixed with

the potassium bromide (KBr, spectroscopic grade) powder at a mass ratio of 1:10 [19]. The obtained mixture was compressed into a small pellet form using a manual hydraulic press and then placed inside a sample holder of the infrared spectrophotometer [20]. The thermal decomposition behavior of the recrystallized samples was analyzed with a thermogravimetric/differential thermal analysis (TG/DTA, Pyris Diamond, Perkin Elmer) instrument. The decomposition characteristics of the recrystallized samples were observed from the obtained TG/DTA profiles at a heating rate of $10\,°C\cdot min^{-1}$ from room temperature to $800\,°C$ under inert ($N_2$) gas at a flow rate of $80\,mL\cdot min^{-1}$. The sample powder was loaded into an alumina pan without a lid and pressing process using the calcined $\alpha$-$Al_2O_3$ as the reference material [21].

## 3. Results and Discussion

### 3.1. Yield and Soluble Percentage

The masses (g) of the recrystallized products, their corresponding yields (%), and the solubility values (%) were investigated, and the results are shown in Table 1. Using 10 g of shell-derived TSP, 5.10, 4.96, and 4.63 g of recrystallized $Ca(H_2PO_4)_2\cdot H_2O$ products were obtained for RCP30, RCP50, and RCP80, respectively. The obtained masses of the products indicated that the yields of the product obtained from the recrystallization process (Table 1) decreased as the temperature used to dissolve the TSP increased, and the product obtained from the dissolved temperature of $30\,°C$ showed the highest recrystallized yield (51.0%). After that, the solubility of the recrystallized products was investigated. From the experimental results, the solubilities of RCP30 and RCP50 were 98.72% and 99.16%, respectively, whereas the solubility of RCP80 (96.63%) was significantly lower than the other samples. According to the previous research, it was reported that the solubility of efficient TSP fertilizer ($Ca(H_2PO_4)_2\cdot H_2O$) is usually more than 85% [22]. This solubility performance is dependent on the fertilizers' sources (i.e., reactant grade used for $Ca(H_2PO_4)_2\cdot H_2O$ preparation) and the operating conditions (i.e., neutral or basic or acidic condition). Consequently, this observation indicated that all recrystallized $Ca(H_2PO_4)_2\cdot H_2O$ obtained in this research are excellent soluble fertilizers in terms of agronomic performance. Additionally, the solubility of the product recrystallized at a low dissolution temperature was higher than the product recrystallized at a higher dissolution temperature, and $50\,°C$ is the suitable dissolution temperature to recrystallize the soluble $Ca(H_2PO_4)_2\cdot H_2O$ fertilizer.

**Table 1.** Mass of recrystallized products, recrystallized yield, and soluble fraction percentage obtained from $Ca(H_2PO_4)_2\cdot H_2O$ samples.

| Samples | Masses of Products (from 10 g TSP)/g | Yields/% | Solubilities/% |
|---------|--------------------------------------|----------|----------------|
| RCP30 | 5.10 | 51.0 | 98.72 |
| RCP50 | 4.96 | 49.6 | 99.16 |
| RCP80 | 4.63 | 46.3 | 96.63 |

When pure $Ca(H_2PO_4)_2\cdot H_2O$ phase was dissolved in the water medium, $H_3PO_4$ and $CaHPO_4$ species were formed [7] according to hydrolyzed reaction, as shown in Equation (2). However, the reactions did not complete because the acid ($H_3PO_4$) was not removed from the system [7]. Therefore, for the solution system without the addition of other chemical reagents, such as a basic reagent, especially sodium hydroxide (NaOH), the equilibrium phenomenon will be obtained, and the $H_3PO_4$ will remain between amorphous $CaHPO_4$ and crystalline $Ca(H_2PO_4)_2\cdot H_2O$. Therefore, when the $Ca(H_2PO_4)_2\cdot H_2O$ product, recrystallized from $Ca^{2+}$-rich oyster waste TSP, was employed as the fertilizer, the largest amount of phosphorus content with the lowest impurities will be valuable for the cultivation process.

### 3.2. Morphology and Chemical Composition (SEM-EDX)

SEM technique was applied to observe the morphological characteristics of the re-crystallized samples. SEM images of RCP30, RCP50, and RCP80 products are shown in Figure 2. The micrograph of RCP30 (Figure 2a) shows many plates of crystal intersperse in different sizes, ranging from around 10–110 μm. Conversely, the morphologies of RCP50 (Figure 2b) and RCP80 (Figure 2c) products present the coagulate plate of crystals with sizes of around 1–90 μm. According to the SEM results, it can be seen that the crystal size of RCP80 (1–50 μm) is smaller than that of the crystal size of RCP50 (5–90 μm). Using the EDX analysis, coupled with the SEM technique, the elemental composition was investigated, and the EDX results, as demonstrated in Table 2, indicate the presence of calcium (Ca), phosphorus (P), and oxygen (O) elements, which are characteristic of the $Ca(H_2PO_4)_2 \cdot H_2O$ recrystallized in the present work. Furthermore, the results of EDX show the presence of potassium (K) and aluminum (Al) elements (slight amount) as the adulterate ions.

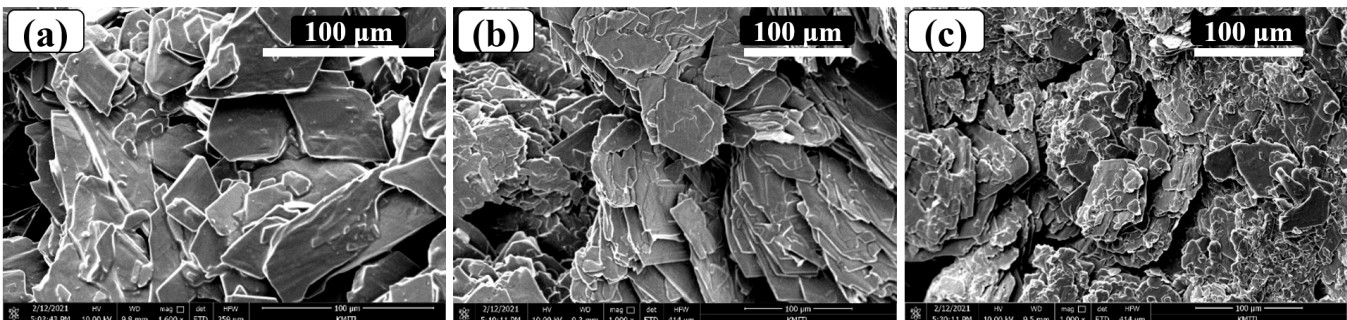

**Figure 2.** Morphological characteristics (scanning electron microscopic (SEM) images by using the gold-coating technique for sample preparation) of $Ca(H_2PO_4)_2 \cdot H_2O$ products recrystallized after dissolving TSP at 30 °C (RCP30 (**a**)), 50 °C (RCP50 (**b**)), and 80 °C (RCP80 (**c**)).

**Table 2.** Chemical elements observed from energy dispersive X-ray (EDX) data of $Ca(H_2PO_4)_2 \cdot H_2O$ recrystallized from oyster-shell-waste TSP samples.

| Elements | RCP30 | RCP50 | RCP80 |
|---|---|---|---|
| Carbon, C | 8.88 | 12.75 | 15.08 |
| Oxygen, O | 55.40 | 47.13 | 48.73 |
| Phosphorus, P | 21.85 | 23.33 | 20.54 |
| Aluminum, Al | – | 0.45 | 1.24 |
| Potassium, K | 0.87 | 1.91 | 1.90 |
| Calcium, Ca | 13.01 | 14.43 | 12.51 |
| **Total** | **100.00** | **100.00** | **100.00** |

### 3.3. X-ray Diffractogram (XRD)

D8 Advance XRD was employed to determine the diffraction pattern of the samples obtained from both non-soluble powders and the recrystallized products. The X-ray diffractograms (or XRD patterns) of non-soluble powders (NSP30, NSP50, and NSP80) samples are displayed in Figure 3a. The diffractogram results indicated that all peaks present the diffraction characteristic of $CaHPO_4 \cdot 2H_2O$ based on the standard ICDD # 72-0713 [23]. From the previous work, the lattice parameters of $CaHPO_4 \cdot 2H_2O$ are *a* = 5.812, *b* = 15.180, and *c* = 6.239 Å with the monoclinic crystal system and space group of *Ia*, whereas the *β* angle equals 116.42° [24]. Its structural layers are comprised of $PO_4$ tetrahedra and $CaO_8$ octahedra. Six O atoms of $CaO_8$ are bound to $PO_4$, and the other two O atoms of $CaO_8$ belong to $H_2O$ [25]. The H is bound to the O atom with the longest P–O bond in the $PO_4$ [26]. Layers within $CaHPO_4 \cdot 2H_2O$ are held together by H bonds and are parallel to the *c*-axis [27]. The H in the two $H_2O$ forms strong bonds with O of $PO_4$; weak H bonds exist between O of $H_2O$ and H of P–OH and also between the molecules of $H_2O$ [28]. The

molecules of $H_2O$ in $CaHPO_4 \cdot 2H_2O$ are not identical, and the H–O–H bond angles of 105.4° and 106.6° were reported by Schofield et al. [25] and Curry and Jones [26]. Furthermore, one molecule of $H_2O$ had the H-bond angles of 167.3° and 165.8°, and another molecule had the angles of 175.7° and 173.3° [26].

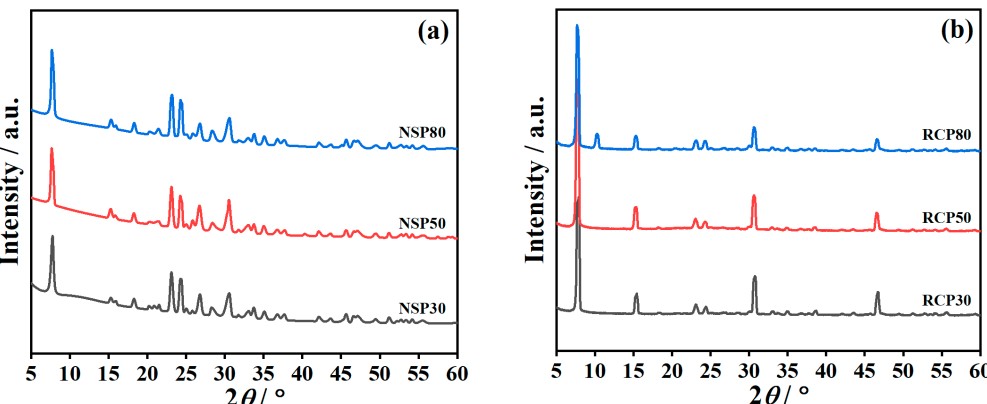

**Figure 3.** X-ray diffraction (XRD) patterns of (**a**) non-soluble powders $CaHPO_4 \cdot 2H_2O$ (NSP30, NSP50, and NSP80) and (**b**) recrystallized $CaHPO_4 \cdot 2H_2O$ (RCP30, RCP 50, and RCP80) compounds obtained in the 2$\theta$ range of 5°–60°.

Figure 3b shows the XRD patterns of recrystallized products (RCP30, RCP50, and RCP80). The peaks with the highest intensities (in 2$\theta$) are observed around 7.8°, 15.4°, 23.1°, 24.4°, 30.7°, and 46.6°. The XRD pattern results indicate and confirm the formation of calcium dihydrogen phosphate monohydrate ($Ca(H_2PO_4)_2 \cdot H_2O$) according to the ICDD # 009-0347, which can be applied as a nutrient, yeast food, leavening agent, hardener, and buffer [3]. According to the previous work, it was reported that $Ca(H_2PO_4)_2 \cdot H_2O$ crystallizes in a triclinic crystal system, and the lattice parameters are $a$ = 6.25 Å, $b$ = 11.89 Å, and $c$ = 5.63 Å with and $\alpha$ = 96.67°, $\beta$ = 114.20°, and $\gamma$ = 92.95° [3]. The crystal structure of $Ca(H_2PO_4)_2 \cdot H_2O$ was well described by Dickens and Bowen [29]. The chains of $CaH_2PO_4^+$ form corrugated layers. The layers of $H_2PO_4^-$ and $H_2O$ were observed in the layers between $CaH_2PO_4^+$ layers. The positions of two H sets differ by approximately 0.28 Å. One H bond of the $H_2O$-donor molecule is apparently bifurcated, whereas other H atoms form normal H bonds, and each site of H atoms was fully occupied [29]. Furthermore, the diffractogram of RCP80 also indicates some residues at 10.26°, which can be attributed to hydrogenated hydrated calcium chloride ($CaClH_2PO_4 \cdot H_2O$) [3] according to the ICDD # 044-000746.

### 3.4. Vibrational Spectroscopy (FTIR)

Infrared adsorptions of non-soluble powders (NSP30, NSP50, and NSP80) are recorded by Spectrum GX FTIR spectrophotometer, and the resulting spectra are shown in Figure 4a. Infrared adsorption frequencies of non-soluble powders are in line with results reported by Tortet et al. [30] and Dosen and Giese [27]. The vibrational characteristics of hydrogen phosphate anion ($HPO_4^{2-}$) and water ($H_2O$) molecules were used to explain the infrared spectra obtained in the present work. The vibrational characteristic modes, sorted from high to low wavenumber (cm$^{-1}$), consisted of O–H stretching of absorbed $H_2O$, O–H stretching of $OH^-$ ion in $HPO_4^{2-}$, (P)O–H stretching of $HPO_4^{2-}$, H–O–H bending and rotation of $H_2O$, H–O–H bending of $H_2O$, very low H–O–H bending of $H_2O$, P–O–H in-plane-bending of $HPO_4^{2-}$, P–O stretching of $HPO_4^{2-}$, P–O(H) stretching of $HPO_4^{2-}$, absorbed $H_2O$, and O–P–O(H) bending of $HPO_4^{2-}$, respectively. Table 3 summarizes the vibrational modes and their corresponding wavenumbers obtained in this research. According to the obtained results, it was confirmed that $CaHPO_4 \cdot 2H_2O$ is the compound found in the non-soluble powder.

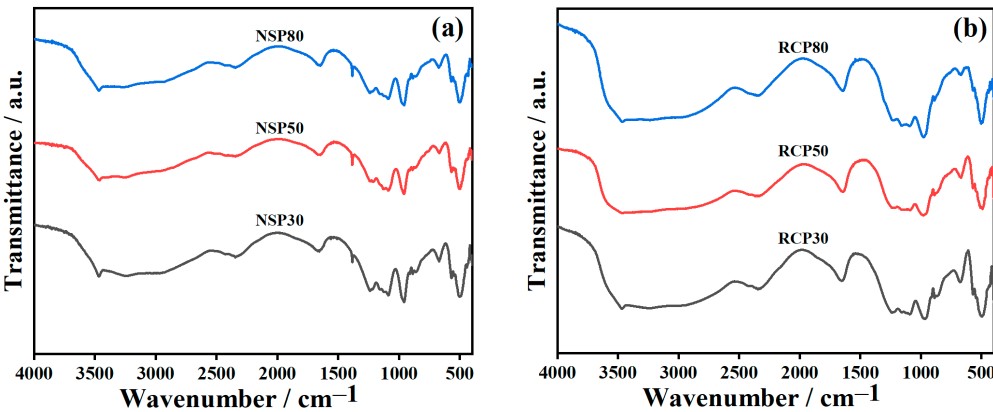

**Figure 4.** Infrared adsorption (FTIR) spectra of (**a**) non-soluble powders $CaHPO_4 \cdot 2H_2O$ (NSP30, NSP50, and NSP80) and (**b**) recrystallized $CaHPO_4 \cdot 2H_2O$ (RCP30, RCP 50, and RCP80) compounds obtained in the wavenumber from 4000–370 $cm^{-1}$.

**Table 3.** Vibrational characteristics (modes) and vibrational positions (wavenumbers/$cm^{-1}$) observed in the non-soluble $CaHPO_4 \cdot 2H_2O$ (NSP30, NSP50, NSP80) and the recrystallized $Ca(H_2PO_4)_2 \cdot H_2O$ (RCP30, RSP50, RCP80) compounds.

| Vibration Modes | Wavenumber (/$cm^{-1}$) of | | | | | |
|---|---|---|---|---|---|---|
| | Non-Soluble $CaHPO_4 \cdot 2H_2O$ Products | | | Recrystallized $Ca(H_2PO_4)_2 \cdot H_2O$ Products | | |
| | NSP30 | NSP50 | NSP80 | RCP30 | RCP50 | RCP80 |
| O–H stretching of absorbed $H_2O$ | 3466 | 3462 | 3465 | 3467 | 3466 | 3462 |
| O–H stretching of $OH^-$ ion in $HPO_4^{2-}$ or $H_2PO_4^-$ | 3244 | 3264 | 3260 | 3248 | 3256 | 3238 |
| (P)O–H stretching of $HPO_4^{2-}$ or $H_2PO_4^-$ | 2970 | 2978 | 2972 | 3006 | 3052 | 3026 |
| H–O–H bending and rotation of $H_2O$ | 2346 | 2346 | 2345 | 2346 | 2346 | 2342 |
| H–O–H bending of $H_2O$ | 1656 | 1647 | 1647 | 1656 | 1642 | 1650 |
| Very low H–O–H bending of $H_2O$ | 1384 | 1385 | 1385 | – | – | – |
| P–O–H in-plane-bending of $HPO_4^{2-}$ or $H_2PO_4^-$ | 1239 | 1238 | 1240 | 1240 | 1235 | 1231 |
| P–O stretching of $HPO_4^{2-}$ | 1155–958 | 1155–959 | 1155–959 | 1159–964 | 1166–980 | 1159–978 |
| P–O(H) stretching of $HPO_4^{2-}$ or $H_2PO_4^-$ | 888 | 890 | 888 | 888 | 888 | 889 |
| Absorbed $H_2O$ | 671 | 670 | 675 | 676 | 669 | 670 |
| O–P–O(H) bending of $HPO_4^{2-}$ or $H_2PO_4^-$ | 569–503 | 570–499 | 570–506 | 570–496 | 571–492 | 570–504 |

Infrared adsorptions of the recrystallized products (RCP30, RCP50, and RCP80) are shown in Figure 4b. By comparison, it is realized that the spectra of RCP30, RCP50, and RCP80 samples are similar. The obtained infrared spectra are mainly characterized based on the vibrational characteristics of dihydrogen phosphate anion ($H_2PO_4^-$) and water ($H_2O$) molecules. The vibrational characteristic modes consisted of O–H stretching of absorbed $H_2O$, O–H stretching of $OH^-$ ion in $H_2PO_4^-$, (P)O–H stretching of $H_2PO_4^-$, H–O–H bending and rotation of $H_2O$, H–O–H bending of $H_2O$, P–O–H in-plane-bending of $H_2PO_4^-$, P–O stretching of $H_2PO_4^-$, P–O(H) stretching of $H_2PO_4^-$, absorbed $H_2O$, and O–P–O(H) bending of $H_2PO_4^-$, respectively. Their wavenumbers are shown in Table 3. Experimental results obtained in this research are consistent with the results reported in the literature [31–33]. The wavenumber at 669–676, 888–889, 964–980, 1080–1095, and 1231–1240 $cm^{-1}$, as demonstrated in Table 3, are the main five vibrational characteristics of $Ca(H_2PO_4)_2 \cdot H_2O$. Consequently, the vibrational characteristics obtained in the present work correspond and confirm the formation of MCPM or $Ca(H_2PO_4)_2 \cdot H_2O$ recrystallized from oyster-shell-derived TSP.

### 3.5. Thermal Analysis (TG/DTA)

The thermal decomposition behaviors of RCP30, RCP50, and RCP80 samples recorded by Pyris Diamond TG/DTA are shown in Figure 5a–c, respectively. The existence of the endothermic peaks located under 200 °C is identified by the loss of physisorbed water (first dehydration, Equation (4)) on the surface of samples, resulting in the formation of calcium dihydrogen phosphate anhydrous ($Ca(H_2PO_4)_2$). The endothermic phenomenon occurred at 234 °C (Figure 5a,b) and 258 °C (Figure 5c), which are related to the dehydroxylation characteristic (polycondensation (orthophosphate $PO_4^{3-} \rightarrow$ pyrophosphate $P_2O_7^{4-}$), second dehydration, Equation (5)) of $Ca(H_2PO_4)_2$ to form calcium dihydrogen pyrophosphate anhydrous ($CaH_2P_2O_7$). Another thermal behavior above 320 °C is the re-polycondensation (pyrophosphate $P_2O_7^{4-} \rightarrow$ metaphosphate $P_2O_6^{2-}$) process (third dehydration, Equation (6)), which corresponds to the formation of calcium metaphosphate anhydrous ($CaP_2O_6$) [34,35]. Consequently, the thermal decomposition equations of the recrystallized $Ca(H_2PO_4)_2 \cdot H_2O$ product are shown below:

First dehydration of physisorbed water:

$$Ca(H_2PO_4)_2 \cdot H_2O(s) \rightarrow Ca(H_2PO_4)_2(s) + H_2O(g) \tag{4}$$

Second dehydration of polycondensation:

$$Ca(H_2PO_4)_2(s) \rightarrow CaH_2P_2O_7(s) + H_2O(g) \tag{5}$$

Third dehydration of re-polycondensation:

$$CaH_2P_2O_7(s) \rightarrow CaP_2O_6(s) + H_2O(g) \tag{6}$$

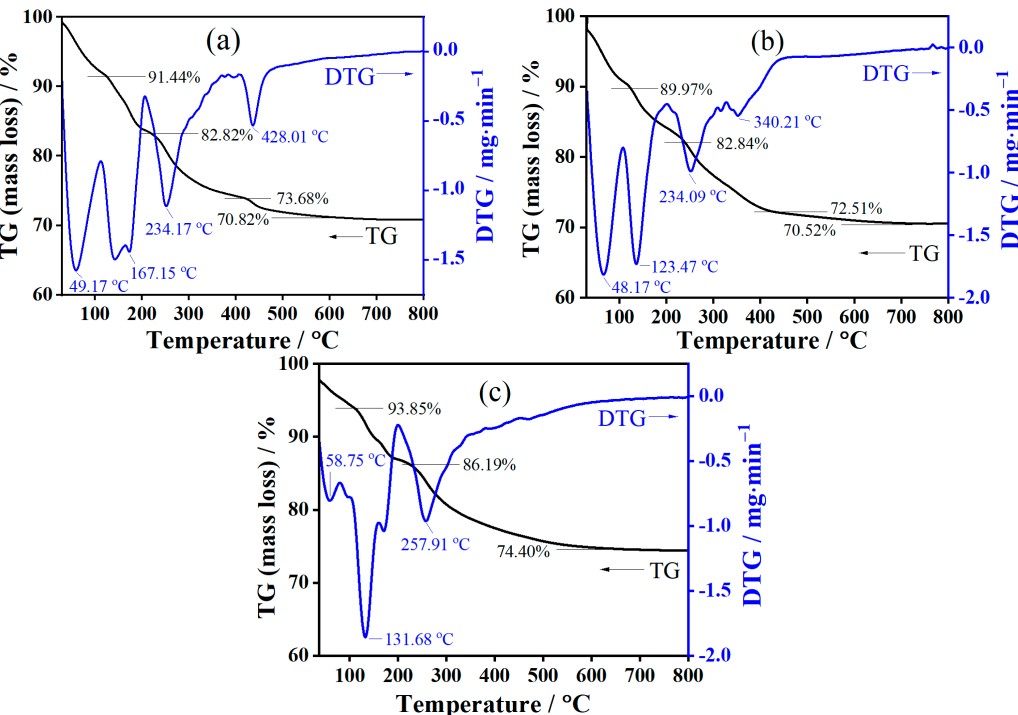

**Figure 5.** Thermal decomposition behaviors (TG/DTA thermograms) of RCP30 (**a**), RCP50 (**b**), and RCP80 (**c**) obtained from room temperature to 800 °C.

Some calcium phosphate compounds showed safety and biological compatibility in organism tissues [34], and $CaP_2O_6$, one of the phosphate-based compounds with a good biocompatibility property, has been used in biomedical fields [34]. Its mechanical values, i.e., high bending strength (~200 Mpa) and low elastic modulus (~45 GPa), are similar to the natural cortical bone [35]. $CaP_2O_6$ is composed of a large porosity (~70%), indicating

that $CaP_2O_6$ with flexible and high-strength properties can be applied as fillers to repair the bone [34]. When it is implanted in bone tissues, the spaces between tissues are filled. The fillers would be embedded by the high-convoluted growth of natural bone into the spaces. Therefore, it can be concluded that the final thermal decomposition product ($CaP_2O_6$) of the recrystallized $Ca(H_2PO_4)_2 \cdot H_2O$ compound obtained in this work can be further applied as the biocompatible material for biomedical fields.

## 4. Conclusions

The recrystallization process was used to prepare a well-soluble $Ca(H_2PO_4)_2 \cdot H_2O$ fertilizer by using synthesized triple superphosphate (TSP) as a precursor, whereas oyster-shell-derived $CaCO_3$ was used as the raw material for TSP preparation. Using this re-crystallized technique, the waste was transformed into valuable material. Under three different dissolved temperatures, RCP30, RCP50, and RCP80 ($Ca(H_2PO_4)_2 \cdot H_2O$) were obtained. The non-soluble products, namely NSP30, NSP50, and NSP80, were also investigated and observed as $CaHPO_4 \cdot 2H_2O$. The RCP30 showed the highest recrystallized yield of 51.0%, whereas the highest soluble percentage of 99.16% was observed from RCP50. Applying the SEM technique, the plate of crystal intersperse in different sizes of RCP30 was observed, whereas the RCP50 and RCP80 show the coagulate crystal plate. The formation and purity of the soluble $Ca(H_2PO_4)_2 \cdot H_2O$ and non-soluble $CaHPO_4 \cdot 2H_2O$ compounds were confirmed by the X-ray diffractograms. Various vibrational modes of $HPO_4^{2-}$, $H_2PO_4^{-}$, and $H_2O$ presented in the crystal structure of the synthesized products observed from the infrared adsorptions also confirm the characteristic of $Ca(H_2PO_4)_2 \cdot H_2O$ and $CaHPO_4 \cdot 2H_2O$. The thermal decomposition of the recrystallized products was investigated. The thermal dehydration of physisorbed water, polycondensation (dehydroxylation), and re-polycondensation processes were observed, and $CaP_2O_6$ was the final thermodecomposed product.

**Author Contributions:** Conceptualization, B.B. and K.C.; Formal analysis, C.S. (Chaowared Seangarun); Investigation, S.S. and C.S. (Chaowared Seangarun); Methodology, S.S., C.S. (Chaowared Seangarun), B.B., N.L., K.C. and W.B.; Resources, S.S., B.B., N.L. and W.B.; Supervision, B.B.; Writing—original draft, B.B. and K.C.; Writing—review & editing, B.B., C.S. (Chuchai Sronsri) and K.C. All authors have read and agreed to the published version of the manuscript.

**Funding:** This work was supported by the Thailand Science Research and Innovation (TSRI) (RE-KRIS/008/64).

**Data Availability Statement:** The data presented in this study are available on request from the corresponding author.

**Acknowledgments:** The authors would like to thank the Scientific Instruments Center KMITL for supporting TGA, FTIR, XRD, and SEM techniques.

**Conflicts of Interest:** The authors declare no conflict of interest.

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
