# Peer review of "Recrystallization of Triple Superphosphate Produced from Oyster Shell Waste for Agronomic Performance and Environmental Issues"

_minerals, doi:10.3390/min12020254_

Round 1
Reviewer 1 Report
In this work, the authors present a potential method for using oyster shell waste as a source for Calcium dihydrogen phosphate monohydrate production. Then, the authors present a potential scheme of such a production and analyze it at three different temperatures 30, 50, and 80 C, presenting corresponding results (like yield).
Overall, as an idea for using oyster shells as a potential source of TSP, the paper presents an interesting concept, that mostly is situated in more economical considerations like profitability and source of the oyster waste vs current demand of the final product. This part is lacking and would be great to learn about in the paper. Potentially, how much of demand can be covered by this type of product sourcing?
Another small question I have is regarding Table 1, where the yields are presented as a function of temperatures. Does this correspond to a phase diagram of the product?
With these answered, I think the paper is good to be published.
Reviewer 2 Report
Please see attachment.

Round 2
Reviewer 2 Report
I recommend this manuscript to publication after small correction:
-please always use a dot, not a comma, in entering values like line 86 page 2